# A3: Android Agent Arena for Mobile GUI Agents

## Abstract

The advancement of Large Language Models (LLMs) and Multimodal Large Language Models (MLLMs) has catalyzed the development of autonomous AI agents. Mobile graphic user interface (GUI) agents, designed to perform tasks on mobile devices, represent a promising application of this technology. However, a significant gap persists in mobile GUI agent evaluation, where existing benchmarks predominantly rely on either static frame assessments such as AndroidControl or offline static apps such as AndroidWorld and thus fail to capture agent performance in dynamic, real-world online mobile applications. To address this gap, we present Android Agent Arena (A3), a novel evaluation system for mobile GUI agents. Unlike existing dynamic evaluation systems, A3 introduces a benchmark of 100 tasks derived from 20 widely-used, online apps across 20 distinct categories from the Google Play Store, ensuring evaluation comprehension. A3 also presents a novel "essential-state" based evaluation method that leverages MLLMs (either commercial or open-source models) as reward models to progressively verify task completion and process achievement. This automated evaluation approach significantly reduces the reliance on manual labor and coding expertise compared with traditional evaluation methods such as in AndroidWorld. Furthermore, A3 includes a toolkit and an evaluator to streamline Android device interaction and facilitate data collection from both human and agent demonstrations. The complete A3 system, including the benchmark and pipeline, will be publicly released to provide a robust foundation for future research and development in mobile GUI agents.

## 1 Introduction

The capabilities of Large Language Models (LLMs) and Multimodal Large Language Models (MLLMs) have advanced significantly, acting as a primary catalyst for innovation in the field of autonomous AI agents. Within this domain, Graphical User Interface (GUI) agents represent a critical area of research. These agents are designed to execute tasks on GUI devices by processing GUI image information directly. Their operational modality is fundamentally visual, distinguishing them from systems that interact with applications via APIs or texts (e.g., HTML and XML strings). This paper concerns mobile GUI agents, which are specifically adapted for the interaction paradigms of the mobile ecosystem. Their prevalence is growing due to their capacity to autonomously execute user commands, which promises to streamline user workflows and reduce the human interaction.

While research into mobile GUI agents has driven significant advancements in models, datasets, and training paradigms, a critical challenge persists in their evaluation. The inherently sequential, multi-step nature of mobile tasks makes it difficult to ascertain final task success. To address this, pioneering benchmarks (Chai et al., 2025; Rawles et al., 2023; Li et al., 2024a)) introduced static frame step evaluation methods. In this paradigm, an agent's performance is assessed based on its ability to predict the correct next action from a single, static screen image. Although this approach offers a valuable measure of single-step accuracy, it fails to simulate real-world interactions. It cannot account for the dynamic state changes of the device or the cascading effects of a single incorrect action, which can lead to complete task failure. Besides, static frame evaluation struggles when tasks can be achieved through different trajectories, for instance, to open one app, some agent chooses to open the app library and then click the app icon while some other tends to directly open the app using ADB commands.

Figure 1: Overview of the Android Agent Arena (A3). The benchmark features tasks from 20 popular mobile applications. These tasks are classified into two categories (Operation and Information Query) and stratified across three difficulty levels (Easy, Medium, and Hard). The A3 evaluation framework supports essential-state based evaluation by two MLLM-based methods, utilizing either a commercial or an open-source finetuned model.

To overcome the limitations of static assessment, the field has moved towards dynamic evaluation systems. These systems employ real or emulated devices, allowing agents to perform predefined tasks in an interactive environment that reflects real-world conditions. Among these, AndroidWorld (Rawles et al., 2025) is a widely-used benchmark, featuring 116 tasks across 15 applications. To facilitate automated evaluation, AndroidWorld utilizes open-source and offline apps. This choice enables a function-based evaluation method where task success is verified by instrumenting the app's source code to check its internal state. However, this reliance on open-source software imposes a significant constraint on the diversity and comprehension of the tasks. Consequently, many common, daily-use app categories that are typically closed-source, such as shopping, travel, and news, are absent, leaving a critical gap in evaluating agent performance on the apps which users interact with most. Furthermore, other dynamic evaluation frameworks (Chen et al., 2024b; Lee et al., 2025; Xing et al., 2024) are also hindered by other issues, including environmental instability, challenges in state resets, high evaluation costs and human labor, and the potential for inaccurate evaluation results.

To address the shortcomings of existing evaluation systems, we propose the **A**ndroid **A**gent **A**rena (A3). Our platform introduces a new benchmark comprising 100 daily-life tasks derived from 20 popular applications, spanning 20 categories from the Google Play Store's top charts, as illustrated in Figure 1. Crucially, this system includes dynamic, online apps in categories such as news, travel, shopping, and etc., which are untestable due to the evaluation constraints of prior systems. The constantly updating nature of these online apps renders traditional function-based evaluation methods inapplicable. Therefore, we also introduce a novel essential-state evaluation method to overcome the issues. This method leverages the capabilities of MLLMs, from large-scale commercial models to finetuned lightweight open-source alternatives, to serve as reward models and autonomously determine task success and it can also evaluate the progress of tasks even the overall object of the task is not achieved. This MLLM-based approach significantly reduces the manual labor and coding expertise required for evaluation while maintaining high levels of accuracy and reliability.

Our contribution can be summarized as follows:

- We introduce the Android Agent Arena (A3), a new benchmark featuring 100 daily-life tasks. These tasks are derived from 20 popular, dynamic, and online mobile apps from 20 categories in Google Play Store top charts, enabling a robust and comprehensive evaluation of agent performance in complex, real-world scenarios previously difficult to assess.

- We propose a novel essential-state evaluation methodology that uses MLLMs to progressively verify task success. This approach utilizes either large-scale commercial models or our own fine-tuned alternatives (A3RM), which serves as a reward model to assess both intermediate progress and the final task objective.

- We open-source the whole pipeline to accelerate research in the field. The pipeline includes modules for streamlined agent execution, data collection (for both agent and human trajectories), and a versatile evaluator module. This module integrates our essential-state methodology and is designed to be flexible for customization for easy adoption by the community.

Table 1: GUI related datasets and benchmarks. The top four rows are GUI agent related datasets, which provide static frame evaluation. The middle six rows are dynamic evaluation systems, which provide different tasks from different apps in different settings. AndroidWorld provides 15 generals apps from non-mainstream open-source F-Droid.

| Name | Eval Mode | # Tasks | # General Apps | Operation | Inf. Query | Online |
|---|---|---|---|---|---|---|
| AITW | static | - | - | ✓ | ✗ | ✗ |
| AndroidControl | static | - | - | ✓ | ✗ | ✗ |
| AMEX | static | - | - | ✓ | ✗ | ✗ |
| GUI-Odyssey | statuc | - | - | ✓ | ✗ | ✗ |
| AndroidArena | dynamic | 221 | 4 | ✓ | ✗ | ✗ |
| Mobile-Env | dynamic | 74 | 5 | ✓ | ✗ | ✗ |
| AndroidWorld | dynamic | 116 | 15 | ✓ | ✓ | ✗ |
| B-Moca | dynamic | 131 | 4 | ✓ | ✗ | ✗ |
| AndroidLab | dynamic | 138 | 5 | ✓ | ✓ | ✗ |
| SPA-bench | dynamic | 170 | 20 | ✓ | ✗ | ✓ |
| A3 (Our) | dynamic | 100 | 20 | ✓ | ✓ | ✓ |

## 2 RELATED WORK

### 2.1 GUI AGENTS

Recent advancements have leveraged the extensive world knowledge and reasoning capabilities of MLLMs for GUI control tasks, aiming to develop more general and autonomous interactive agents (Liu et al., 2025a; Wang et al., 2025b; Hu et al., 2025). A prominent approach employs general-purpose multimodal models as GUI controllers, such as GPT-4v and Qwen2.5-VL, integrating visual and textual inputs to plan and execute tasks (Zheng et al., 2024; Bai et al., 2025). An alternative line of research focuses on lightweight and domain-specific models that incorporate GUI knowledge to improve efficiency and task performance. For instance, CogAgent (Hong et al., 2024) enhances GUI task performance via a high-resolution cross-modal fusion module; SphAgent (Chai et al., 2025) leverages element functionalities for fine-grained screen and component understanding; UI-TARS (Qin et al., 2025) finetunes on additional reasoning and GUI perception data; GUI-Owl (Wang et al., 2025a) applies layout-guided contrastive learning for rapid GUI grounding transfer. Several recent works also explore combining multimodal perception with chain-of-thought reasoning and self-reflection (Wu et al., 2025) for autonomous operation in complex environments. AppAgent (Li et al., 2024b) employs multi-round CoT reasoning with execution feedback for closed-loop control, while GUI Critic (Wanyan et al., 2025) introduces pre-action critique and self-reflection modules, combined with S-GRPO reinforcement finetuning and tool interaction correction, highlighting its potential for error prevention in dynamic GUI environments.

### 2.2 GUI AGENT BENCHMARKS

Early evaluations of GUI agents primarily relied on static benchmarks where an agent predicts the next action from a single screenshot. Seminal works such as AITW (Rawles et al., 2023), AMEX (Chai et al., 2025), and AndroidControl (Li et al., 2024a) focused on single-step action accuracy via element or coordinate matching. While valuable for component-level assessment, this static paradigm fundamentally fails to capture an agent's ability to perform sequential tasks in dynamic, interactive environments. This limitation motivated the development of the dynamic evaluation frameworks that are the focus of our work.

However, existing dynamic benchmarks exhibit significant constraints in their task and application design, limiting their realism and scope. Several systems are restricted by task simplicity and diversity, such as Mobile-Env (Zhang et al., 2023) and B-Moca (Lee et al., 2025). Others are constrained by their app selection; AndroidArena (Tong et al., 2024), for instance, focuses on Google and system apps that are often manageable via APIs, which fails to test an agent's generalizability to the broader third-party ecosystem. Similarly, prominent benchmarks like AndroidWorld (Rawles et al., 2025) and AndroidLab (Xu et al., 2024) are restricted to offline or open-source apps. The UIs and functionalities of these apps often deviate from mainstream commercial applications, and by design, they lack

the stochastic events common in real-world usage—such as pop-up advertisements and dynamic content updates—that are critical for testing agent robustness. While SPA-bench (Chen et al., 2024b) attempts to incorporate online apps, it suffers from severe environmental instability and difficult resets, hindering reliable experimentation. A more foundational issue across many of these benchmarks is the reliance on simplistic evaluation methodologies. Information query success is often determined by matching predefined answers (Xu et al., 2024), and operational success by exact state matching (Rawles et al., 2025). These rigid, programmatic checks fail to capture the nuances of task completion in dynamic environments. This highlights a critical need for a benchmark with ecologically valid tasks coupled with a more flexible, semantically-aware evaluation framework. The overall statistics of existing benchmarks are listed in Table 1.

## 3 ANDROID AGENT ARENA (A3)

### 3.1 TASKS & APPS

The Android Agent Arena (A3) is a benchmark composed of 100 representative tasks sourced from 20 popular, dynamic apps spanning 20 distinct categories from the Google Play Store top charts and averaging 115 million downloads each. To provide a structured evaluation, tasks are classified along two primary axes. Based on their objective, they are designated as either Operation (requiring a sequence of state-changing actions) or Information Query (which additionally requires the agent to extract information to answer a question). The difficulty of each task is quantitatively defined by the number of steps ($N$) required by a human expert: Easy ($N < 7$), Medium ($7 \leq N \leq 11$), and Hard ($N > 11$).

Our application selection process is designed to overcome the critical limitations of prior benchmarks, which often rely on offline, open-source apps with limited task diversity (Xu et al., 2024; Rawles et al., 2025) or use online apps that require significant manual intervention for environment resets (Chen et al., 2024b). We undertake a rigorous curation process, analyzing over 100 applications from Google Play Store top charts. Our final selection of 20 applications (detailed in Appendix A.1) is guided by two primary criteria: (1) maximizing coverage of categories and (2) ensuring minimal login or environment reset, which facilitates accessible and repeatable experimentation. This careful curation yields the key advantage of A3: resolving the trade-off between ecological validity and experimental reproducibility. It creates a unique testing environment that supports dynamic, online app states while still allowing for reliable and programmatic resets to a consistent initial environment. This enables a robust Mobile GUI agent evaluation on the types of complex, dynamic tasks that users perform daily, all within a stable and reproducible framework.

### 3.2 ESSENTIAL-STATE EVALUATION

Existing evaluation methodologies for mobile agents often rely on metrics that lack sufficient granularity and flexibility for in-depth analysis. The widely-used AndroidWorld benchmark (Rawles et al., 2025), for instance, primarily employs a binary Success Rate (SR), calculated as the ratio of successful tasks to the total ($N_{success}/N_{total}$). While straightforward, this coarse-grained metric provides no insight into partial progress or specific failure modes, and it cannot differentiate the capabilities of agents that achieve the same final score. To address this, AndroidLab (Xu et al., 2024) introduced an additional, more granular sub-goal success rate. While this approach offers a more detailed view by tracking intermediate steps, its implementation has significant limitations. The sub-goals are pre-defined in a rigid JSON format (e.g., *Phone: 12345678* ) for the evaluation function and are only applicable to certain operation tasks.

To address the need for a granular yet flexible metric, we introduce the core of A3's evaluation framework: the "essential-state" evaluation method. We define an essential-state as a critical milestone or observable outcome that must be achieved for a task to be considered successful. Instead of assessing a single, final task outcome, our method decomposes each task into a sequence of these verifiable states. This approach differs fundamentally from the rigid sub-goals used in systems like AndroidLab (Xu et al., 2024). While sub-goals are often tied to specific key-value pairs, essential-states are defined by higher-level outcomes, making them more resilient to variations in UI or agent execution paths. This flexibility is crucial for evaluating performance on dynamic apps and for information query tasks. For instance, for the task *"Search 'marvel comics' in Pinterest. Who is*

Figure 2: Illustration of the sliding window mechanism for essential-state evaluation. A window of a predefined size and interval traverses the agent's interaction trajectory. At each position, the screen frames within the window are passed to an MLLM-based judge. The judge then assesses whether any of the task's essential states have been achieved within that segment. This process repeats until the entire trajectory has been evaluated.

*the author of the first pin?"*, we define three essential states: (1) 'marvel comics' is searched, (2) First pin is selected, and (3) Author of the pin is answered. The latter two states represent flexible achievements that are infeasible to encode with AndroidLab sub-goals.

For essential-state evaluation metric, we define $ESAR = N_{AES}/N_{TES}$, where $ESAR$ is the **E**ssential-**S**tate **A**chieved **R**ate, $N_{AES}$ is the number of achieved essential-states and $N_{TES}$ is the total number of essential-states. Overall task success is then determined by the successful completion of all its associated essential states. We formalize this as:

$$A_i = \begin{cases} 1, & \text{if } eval(ES_j) = 1 \text{ for } \forall j \in [0, ..., N_{ES}] \\ 0, & \text{otherwise} \end{cases}$$

where $A_i$ is the success status of task $i$, $ES_j$ is the $j$-th essential-state for the task, $N_{ES}$ is the total number of the essential-states for task $i$ and $eval(\cdot)$ is the method that verifies the achievement of an essential-state. Thus, the complex problem of end-to-end task evaluation is simplified into a series of more manageable essential-state assessments.

To implement the essential-state evaluation, we process the agent's full interaction trajectory using a sliding window mechanism. A window of a predefined size traverses the sequence of screen frames from start to finish, advancing at a specified interval. At each position, the contents of the window are submitted to our MLLM-based evaluator, which judges whether any of the essential-states are satisfied within that specific segment. This evaluation process is illustrated in Figure 2.

## 3.3 MLLM-BASED JUDGMENT

A critical challenge in our framework is how to verify the achievement of each essential state. Traditional function-based methods, which rely on parsing UI structures like the XML or View Hierarchy, are not suitable for this task of dynamic, constantly updating apps. These methods often struggle for two primary reasons: (1) many commercial apps feature mis-aligned or inconsistent UI trees that are difficult to parse reliably, and (2) they suffer from a semantic gap, as many essential states are defined by high-level concepts (e.g., *"the first product is selected"*) that cannot be verified by simply checking for a specific widget ID or text string in an constantly updating environment.

To overcome these limitations, we leverage the sophisticated visual and language understanding capabilities of modern MLLMs. Inspired by recent work on using MLLMs as evaluators (Chen et al., 2024a), we employ an MLLM to act as the core judgment mechanism for A3. Our approach, however, refines how the MLLM is applied. Prior work like SPA-bench (Chen et al., 2024b) pioneered the

Figure 3: Illustration of A3 pipeline. AITK has a translator to convert agent actions to a unified action space and a controller to interact with the device. AITK also gets the state of device at each step and save the data. After the task execution, m-evaluator traverses the trajectories and output the evaluation results.

use of an MLLM to directly evaluate an entire, multi-step task trajectory based on the screenshots combination. This naive end-to-end judgment is a complex reasoning task that resulted in an accuracy of approximately only 80% in experiment evaluations, which is unreliable. In contrast, our essential-state evaluation method is perfectly designed to address this challenge by decomposing the overall task into a series of simpler, more discrete verification steps of essential-states. This simplifies the reasoning required of the MLLM, as it makes a focused judgment on a single, much simpler milestone rather than a long sequence, thereby aiming for higher and more reliable evaluation accuracy.

**Commercial MLLMs**  The advanced capabilities of state-of-the-art commercial MLLMs, such as Gemini-2.5-pro, make them highly effective for the role of an automated evaluator. Their robust performance in visual comprehension and instruction following provides a reliable evaluation for judging whether an essential-state has been achieved within a given set of screen frames. Though commercial MLLMs performs good as an evaluator, a significant practical challenge associated with these commercial models is the monetary cost of their API services, as each evaluation call incurs a fee. This creates a critical trade-off between evaluation granularity and cost. Therefore, to ensure our framework is both accurate and economically viable, we need to find a optimal sliding window parameters, specifically the window size and interval size, to minimize the total number of API calls without compromising evaluation fidelity (detailed in Section 4.1).

**Open-source MLLMs**  While optimizing the sliding window parameters mitigates the expense of using commercial APIs, a persistent monetary cost remains for each evaluation. To address this fundamental challenge of cost and accessibility, we propose a lightweight, open-source alternative that eliminates API fees and allows for local deployment, thereby providing researchers with greater control and transparency. To this end, we introduce the A3RM (Android Agent Arena Reward Model), a specialized reward model fine-tuned from MiMo-VL-7B (Xiaomi, 2025). We selected this base model for its strong demonstrated capabilities in GUI grounding and visual understanding, making it an ideal foundation for our evaluator. In summary, A3RM serves as an accurate, cost-effective, and transparent judge for essential-state verification, aiming to democratize robust agent evaluation and enable broader, more accessible research.

### 3.4 A3 PIPELINE SUITE

To accompany our benchmark, we introduce a comprehensive, open-source software pipeline, as illustrated in Figure 3. The core of this suite is the AITK (Android Interaction Toolkit), a lightweight and customizable framework for managing agent-device interaction, data collection, and task execution. AITK is mainly designed for model-based agents and it features a plug-in architecture

Table 2: Sliding window and interval size study for essential-state evaluation. The highest accuracy is in bold. The average cost is computed by the total API cost of three MLLMs over 30 tasks.

| Window Size | Interval Size | Essential-State Acc | Task Acc | Average Cost ($) |
|---|---|---|---|---|
| 2 | 1 | 0.94 | **0.91** | 0.196 |
|   | 2 | 0.88 | 0.86 | 0.099 |
| 3 | 1 | **0.95** | **0.91** | 0.195 |
|   | 2 | 0.94 | 0.90 | 0.098 |
| 4 | 2 | **0.95** | **0.91** | 0.098 |
|   | 3 | **0.95** | 0.90 | 0.071 |
| 5 | 2 | 0.93 | 0.89 | 0.097 |
|   | 3 | 0.92 | 0.89 | 0.069 |
| 6 | 3 | 0.91 | 0.88 | 0.070 |
|   | 4 | 0.89 | 0.87 | 0.050 |

for integrating custom agents and tasks, and supports parallel execution across multiple devices for accelerated testing. Another key component of the suite is a module M-Evaluator, which implements our essential-state evaluation methods. This evaluator is designed to be agent-agnostic, allowing it to be applied to any standard trajectory data, independent of the agent and framework that produced it. The complete pipeline, including our human annotation tools, will be open-sourced to foster reproducibility and advance future research.

## 4 EXPERIMENTS

### 4.1 SLIDING WINDOW AND INTERVAL STUDY

We conduct experiments on the sliding window size and interval size (number of steps) for the commercial MLLMs (Gemini) evaluation to identify the optimal parameters to minimize the number of API calls without compromising evaluation fidelity. Table 2 summarizes the evaluation results across different sliding window sizes and interval settings. Specifically, the sliding window size ranges from 2 to 6, with interval sizes set to half of the window size. From the results, we observe that sliding window sizes of 3 and 4 yield the highest evaluation accuracies for both essential states and overall task performance. Among these, the configuration with a sliding window size of 4 and an interval size of 2 achieves the lowest average computational cost, making it the preferred setting for the evaluation method.

Further analysis of our results reveals a distinct trade-off in the selection of the sliding window size, with the MLLM evaluator's performance degrading at both small and large extremes. When a small window is combined with a non-overlapping stride (e.g., size of 2, stride of 2), evaluation accuracy suffers. This occurs because a critical state transition can be split across the boundary of two consecutive windows. For example, an agent's action and its immediate on-screen result might fall into separate windows, depriving the MLLM of the necessary context to make a correct judgment. Conversely, when the window size becomes too large (e.g., greater than 4), performance degrades due to loss of visual fidelity. To be processed, frames in the window are concatenated into a single composite image. With larger window sizes, each frame is significantly downscaled, causing a severe loss of resolution that can render small text and UI elements illegible. This image compression impairs the MLLM's ability to accurately identify essential states, leading to misjudgments.

### 4.2 A3 REWARD MODEL (A3RM)

We developed the A3RM (Android Agent Arena Reward Model) as a deployable alternative reward model. We selected MiMo-VL-7B (Xiaomi, 2025) as the base model, given its strong performance on GUI grounding and screen understanding tasks. Interestingly, while larger window sizes were optimal for commercial models, our preliminary experiments revealed that a smaller window size of 2 yielded the best performance for the 7B-parameter MiMo-VL model. This is likely due to its more limited context capacity compared to larger commercial models. Consequently, we adopted a window size of 2 and interval size of 1 for all A3RM data collection and training.

Table 3: A3RM evaluation metric. Ess. State stands for essential-state.

| Model | Ess. State Precision | Ess. State Recall | Ess. State F1 | Task Precision | Task Recall | Task F1 |
|---|---|---|---|---|---|---|
| MiMo-VL-7B | 73.0 | 75.3 | 74.2 | 67.8 | 76.0 | 71.7 |
| Gemini-2.5-pro | 87.3 | 96.3 | 91.5 | 85.7 | 96.0 | 90.6 |
| A3RM (7B) | 89.2 | 90.0 | 89.6 | 91.7 | 88.0 | 89.9 |

**Data Collection**  We curated a high-quality dataset by collecting an average of three distinct successful human trajectories for each of our 100 tasks to ensure coverage of multiple valid solution paths. Human annotators identified the precise windows where essential states were achieved; these constitute our positive samples, while all other windows from these expert trajectories serve as negative samples. To enrich this dataset with diverse failure cases, we also employed an automated negative mining strategy. We leveraged the high-recall, low-precision nature of the Gemini-based evaluator by running it over agent-generated trajectories, allowing us to efficiently identify and label additional negative samples.

**Training**  We finetuned MiMo-VL-7B using the dynamic sampling policy optimization (DAPO) algorithm (Yu et al., 2025) on our curated dataset. Given the natural class imbalance in our data (far more negative windows than positive ones), we employed an oversampling strategy for the positive samples in each training epoch to prevent the model from collapsing to a trivial solution of always predicting a negative outcome. More training details are in Appendix A.4.

**Results**  The final performance of A3RM is presented in Table 3, where it is compared against both the zero-shot MiMo-VL-7B baseline and the Gemini-based evaluator on a held-out test set. The results demonstrate that A3RM's performance is highly competitive with the much larger and more costly commercial model. A deeper analysis reveals that A3RM achieves higher precision at the cost of slightly lower recall than the Gemini evaluator. This trade-off indicates that A3RM operates with stricter, more human-aligned judgment criteria, making it less prone to false positives. Ultimately, this outcome validates our approach of creating a specialized and accurate evaluator that provides a reliable and accessible tool for the community. More analysis is in Appendix A.5.1 .

## 4.3   A3 BENCHMARK

To establish a comprehensive baseline and demonstrate the challenges posed by our benchmark, we evaluated a diverse suite of mobile GUI agents. We select six single-model agents: Qwen2.5-VL (Bai et al., 2025), UI-TARS-1.5 (Qin et al., 2025), UI-Genie (Xiao et al., 2025), UI-Venus (Gu et al., 2025), InfiGUI-R1 (Liu et al., 2025b), and GUI-OWL (Ye et al., 2025). We also included two agent frameworks: T3A (Rawles et al., 2025) with Gemini-2.5-pro and Mobile-Use (Li et al., 2025) with Qwen2.5-VL-7B. Each agent was evaluated on the A3 benchmark using its official, publicly available prompts and settings to ensure reproducibility. The detailed results are presented in Table 4.

The results reveal that the A3 benchmark poses a significant challenge for current single-model mobile GUI agents. The overall task Success Rate (SR) is low overall, with the top-performing open-source agent, InfiGUI-R1, achieving just 29.0% under Gemini evaluation and 27.0% under A3RM evaluation. This stands in contrast to the higher success rates (often exceeding 60%) reported on benchmarks like AndroidWorld. We argue that this performance drop demonstrates a clear difficulty gap and validates A3 as a more realistic benchmark that effectively captures the complexities of interacting with dynamic, real-world applications. Furthermore, the results underscore the critical role of agent frameworks and the choice of the underlying foundation model. We observe that structured frameworks can enhance agent performance, as in Mobile-Use with Qwen2.5-VL, but the capabilities of the base model appear to be a dominant factor. An example is the T3A framework, when equipped with a powerful commercial MLLM like Gemini-2.5-pro as its backbone, this old framework still achieves a notable top-tier success rate on our benchmark.

Our proposed Essential State Achieved Rate (ESAR) provides a more nuanced understanding of agent capabilities beyond the binary SR metric. A consistent trend across all agents is that the ESAR is substantially higher than the corresponding SR. For example, while InfiGUI-R1 achieves a 27.0% SR, its ESAR of 52.1% indicates that it successfully completed over half of all required intermediate steps across all tasks. This discrepancy highlights a common failure mode: agents are often capable of

Table 4: A3 benchmark performance results. Results for eight agents, evaluated by both a commercial MLLM (Gemini) and our open-source A3RM. We report the Task Success Rate (SR) and Essential State Achieved Rate (ESAR), with detailed breakdowns for each task category and difficulty level.

| Agent | Evaluator | Metric | Easy | Medium | Hard | Operation | Inf. Query | Overall |
|---|---|---|---|---|---|---|---|---|
| UI-TARS-1.5-7B | Gemini | SR | 14.3 | 5.0 | 4.0 | 9.7 | 3.5 | 8 |
| | | ESAR | 29.4 | 17.2 | 9.3 | 20.2 | 13.2 | 18.1 |
| | A3RM | SR | 11.4 | 2.5 | 0.0 | 6.9 | 0.0 | 5 |
| | | ESAR | 25.9 | 13.3 | 9.4 | 17.4 | 11.0 | 15.5 |
| UI-Venus-7B | Gemini | SR | 31.4 | 17.5 | 12.0 | 22.2 | 17.8 | 21 |
| | | ESAR | 47.1 | 27.3 | 31.2 | 36.2 | 28.6 | 34.0 |
| | A3RM | SR | 28.5 | 15.0 | 16.0 | 20.8 | 17.8 | 20 |
| | | ESAR | 41.2 | 29.7 | 27.1 | 33.9 | 27.5 | 32.0 |
| UI-Genie-7B | Gemini | SR | 28.6 | 15.0 | 0.0 | 20.8 | 3.6 | 16 |
| | | ESAR | 47.1 | 28.1 | 31.3 | 37.2 | 27.5 | 34.3 |
| | A3RM | SR | 25.8 | 10.0 | 0.0 | 16.7 | 3.6 | 13 |
| | | ESAR | 41.2 | 25.8 | 32.3 | 34.9 | 25.3 | 32.1 |
| Qwen2.5-VL-7B | Gemini | SR | 8.6 | 2.5 | 4.0 | 6.9 | 0.0 | 5 |
| | | ESAR | 25.9 | 14.8 | 26.0 | 22.0 | 19.8 | 21.4 |
| | A3RM | SR | 5.7 | 0.0 | 4.0 | 4.2 | 0.0 | 3 |
| | | ESAR | 10.5 | 10.9 | 21.8 | 15.6 | 11.0 | 14.2 |
| InfiGUI-R1-7B | Gemini | SR | 40.0 | 32.5 | 8.0 | 36.1 | 10.7 | 29 |
| | | ESAR | 58.8 | 53.1 | 52.1 | 56.4 | 49.5 | 54.4 |
| | A3RM | SR | 34.3 | 30.0 | 12.0 | 34.7 | 7.1 | 27 |
| | | ESAR | 55.3 | 50.8 | 51.0 | 54.6 | 46.2 | 52.1 |
| GUI-OWL-7B | Gemini | SR | 25.7 | 5.0 | 4.0 | 15.3 | 3.6 | 12 |
| | | ESAR | 51.7 | 27.3 | 21.8 | 36.2 | 23.1 | 32.4 |
| | A3RM | SR | 22.8 | 5.0 | 0.0 | 13.9 | 0.0 | 11 |
| | | ESAR | 44.7 | 25.8 | 20.8 | 33.9 | 18.7 | 29.5 |
| Mobile-Use + Qwen2.5-VL-7B | Gemini | SR | 34.3 | 15.0 | 4.0 | 25.0 | 3.6 | 19 |
| | | ESAR | 47.1 | 43.0 | 30.2 | 43.1 | 33.0 | 40.1 |
| | A3RM | SR | 34.3 | 10.0 | 0.0 | 22.2 | 0.0 | 16 |
| | | ESAR | 48.2 | 39.9 | 31.3 | 43.1 | 30.8 | 39.5 |
| T3A + Gemini-2.5-pro | Gemini | SR | 62.8 | 57.5 | 52.0 | 62.5 | 46.4 | 58 |
| | | ESAR | 71.76 | 70.3 | 65.6 | 76.6 | 51.6 | 69.3 |
| | A3RM | SR | 57.1 | 55.0 | 44.0 | 55.6 | 46.6 | 53 |
| | | ESAR | 68.2 | 67.9 | 62.5 | 72.9 | 50.5 | 66.4 |

partial task execution but lack the robustness to complete the full sequence of essential states required for success. Besides, the results show a consistent trend when comparing the two MLLM-based evaluators. The Gemini-based evaluator tends to assign slightly higher scores for both SR and ESAR than our A3RM. We hypothesize this is because A3RM, having been fine-tuned on precise human annotations, acts as a stricter and more discerning judge. This aligns with our earlier observation that Gemini exhibits high recall but lower precision, occasionally accepting outcomes that a human annotator would probably reject. While both evaluators capture similar relative performance rankings among agents, A3RM provides a rigorous, cost-effective, and human-aligned standard for evaluation. More analysis and case studies are in Appendix A.5.2.

## 5 CONCLUSION

The evaluation of mobile GUI agents has been hindered by a lack of benchmarks that reflect real-world user scenarios. In this work, we presented Android Agent Arena (A3) as a comprehensive solution. A3 contributes not only a diverse benchmark of tasks on popular online apps but also a novel and flexible "essential-state" evaluation method using MLLMs. By also developing and releasing A3RM, a powerful deployable evaluator that eliminates cost barriers, we are making reliable MLLM evaluation accessible to all researchers. We are releasing the entire A3 ecosystem as an open-source platform to establish a new, more rigorous standard for benchmarking and to catalyze the development of more capable and reliable mobile agents that can operate effectively "in the wild."

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

# A APPENDIX

## A.1 APP AND CATEGORY LIST

Table 5: List of apps and corresponding categories.

| App | Category | App | Category | App | Category |
|---|---|---|---|---|---|
| CNN | News | TripAdvisor | Events | Amazon | Shopping |
| Omio | Travel | Bluecoins | Finance | Tasks | Productivity |
| N Calendar | Calendar | File | Tools | eboox | Read |
| YouTube | Video | Home Workout | Fitness | Pinterest | Lifestyle |
| Chrome | Browser | CityMapper | Navigation | Joytify | Music |
| Joplin | Notes | Wikipedia | Entertainment | Weather Forecast | Weather |
| Supercook | Food | Gmail | Business | | |

## A.2 LLM USAGE

Except for the MLLM evaluation method, we use LLM (Gemini-2.5-pro) to help paper polishing.

## A.3 TASK DESIGN

For task design, we adopt a stringent filtering and validation procedure to ensure consistent step granularity across all tasks, thereby avoiding adverse effects of overly coarse steps on the evaluation. All essential states are manually created, which not only enforces uniform granularity but also guarantees full coverage of the overall task objective. This design ensures that every essential state can be consistently captured by adjacent sliding windows of size greater than or equal to two, thereby preserving the validity of the evaluation method.

## A.4 A3RM TRAINING

For the training of A3RM, we first curated a dataset by manually collecting 8,241 step-wise samples from expert trajectories, comprising 981 positive and 7,260 negative instances. This set was further augmented with 1,083 challenging negative samples identified from agent trajectories and labeled via Gemini-2.5-pro. To address the significant class imbalance in the resulting dataset, we oversampled the positive class by a factor of four. Using this more balanced data, we fine-tuned the MiMo-VL-7B base model with the DAPO algorithm (Yu et al., 2025) on the EasyR1 framework (Zheng et al., 2025). The training was conducted on a cluster of 16 NVIDIA L40S GPUs and completed in two days.

## A.5 BENCHMARK ANALYSIS

### A.5.1 MLLM-BASED EVALUATION

An analysis of evaluation errors reveals that the primary failure modes for both Gemini and A3RM stem from two key issues: visual hallucination and semantic misinterpretation. While powerful, these MLLM-based judges are not infallible, and understanding their limitations is crucial.

The most common error is a form of action-based hallucination, where the MLLM appears to be overly influenced by the agent's described action rather than the resulting visual evidence as illustrated in Figure 4. For example, an agent may execute a click on a specific button intended to complete an essential state. If the UI fails to change in the subsequent screenshot, indicating an unsuccessful interaction, the MLLM evaluator might still incorrectly infer that the action succeeded and mark the state as complete. In these cases, the model hallucinates the expected outcome based on the action's intent, overlooking the contradictory visual proof.

A second category of error is semantic misinterpretation, where the MLLM correctly perceives the visual elements but misunderstands their functional context as illustrated in Figure 5. For instance, if a task's essential state is setting a task title to "Lunch" and setting the cost to "15", an agent might erroneously set the title to "15" after the title is correctly set to "Lunch". The MLLM, however, would

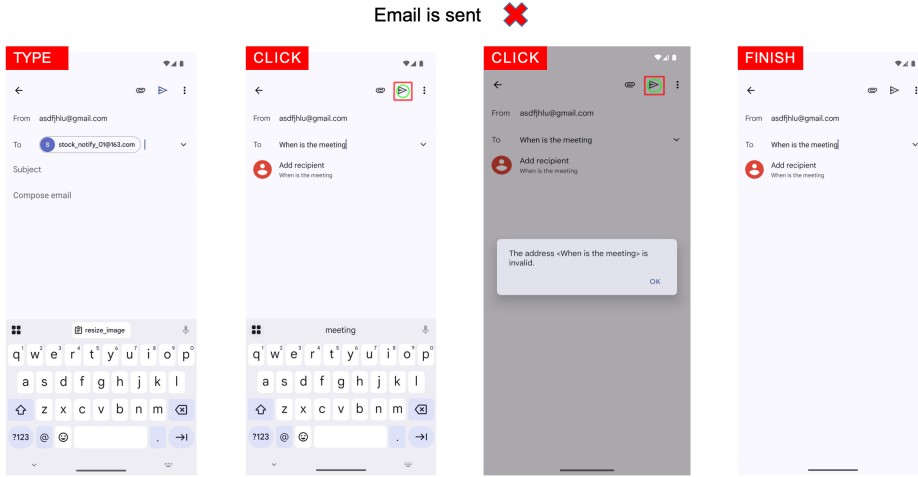

Figure 4: Example of hallucination in MLLM judgments. The email is not set due to invalid address. But the reward model thinks the click on the sending icon achieves the essential-state.

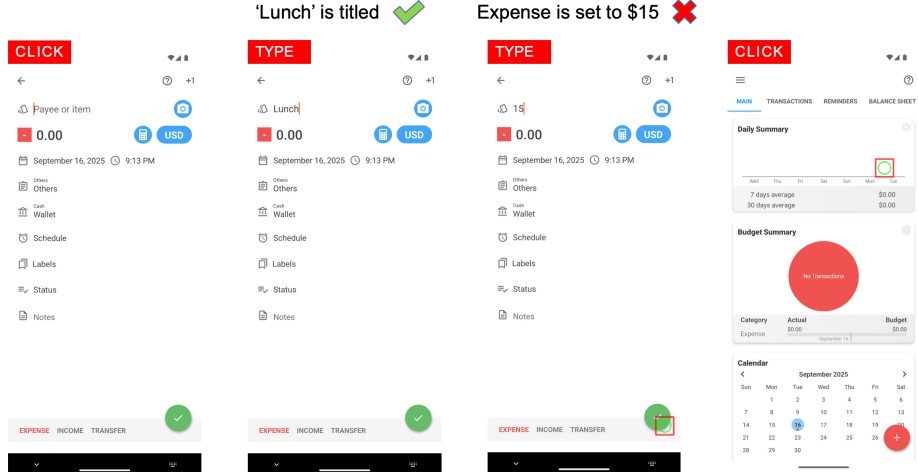

Figure 5: Example of screen misunderstanding in MLLM judgments. In this case, the expense is not set to 15 but the reward model thinks it is set to 15.

recognize the text entry setting to "15", and then incorrectly verify the essential state as achieved, failing to distinguish the critical semantic difference between a "title" and a "cost."

These failure modes underscore the ongoing challenges in grounding abstract task goals to concrete visual evidence and highlight the need for continued advancements in the fine-grained comprehension capabilities of MLLMs for truly robust automated evaluation.

### A.5.2 CASE STUDY

Through observation and analysis of the eight agents performance on A3 benchmark, we noticed some representative and remarkable cases where researchers would like to solve in the future study.

**Progress Unawareness** A critical failure mode observed across most model-based agents is Progress Unawareness, a fundamental inability to track their position within a task sequence. This limitation typically manifests in two ways:

- **Redundant Actions**: Agents often fail to recognize that a required step has already been completed. We observed numerous instances where an agent would get stuck in a loop, repeatedly attempting to achieve an essential state that was already satisfied, even when its action history was provided in the prompt (Figure 6). This suggests a failure to effectively parse its own operational history.
- **Failure to Terminate**: A related issue is the inability to recognize overall task completion. Many agents, despite having successfully achieved all essential states, do not issue a stop action. Instead, they continue to perform irrelevant operations until the AITK framework terminates them for exceeding the maximum step limit, as illustrated in Figure 7.

This lack of progress awareness points to a deeper deficiency in the agents' planning and state-tracking capabilities. An agent that cannot reliably determine what it has already done or recognize when its final goal has been met will struggle with complex, multi-step tasks. Improving this self-awareness and goal recognition is therefore a crucial direction for future research in developing more robust and efficient GUI agents.

**Screen misunderstanding**    Another commonly observed failure mode is a lack of Screen Comprehension, where an agent fails to accurately ground its intended action to the correct visual element on the screen. This issue, which primarily affects `click` actions, manifests in two distinct ways:

- **Incorrect Element Identification:** In some cases, the agent's reasoning is sound (e.g., "click the menu button") but it visually misidentifies the target, instead clicking a different element like a profile icon. This represents a failure in high-level visual recognition, as shown in Figure 8 left.
- **Inaccurate Coordinate Localization:** In other instances, the agent correctly identifies the target element but fails to predict its precise coordinates. This results in a "shifted click" that lands on an adjacent area instead of the intended element, as illustrated in Figure 8 right. This points to a limitation in fine-grained spatial reasoning.

Both error types highlight a core challenge for current models in achieving reliable, fine-grained visual grounding on dense and complex user interfaces.

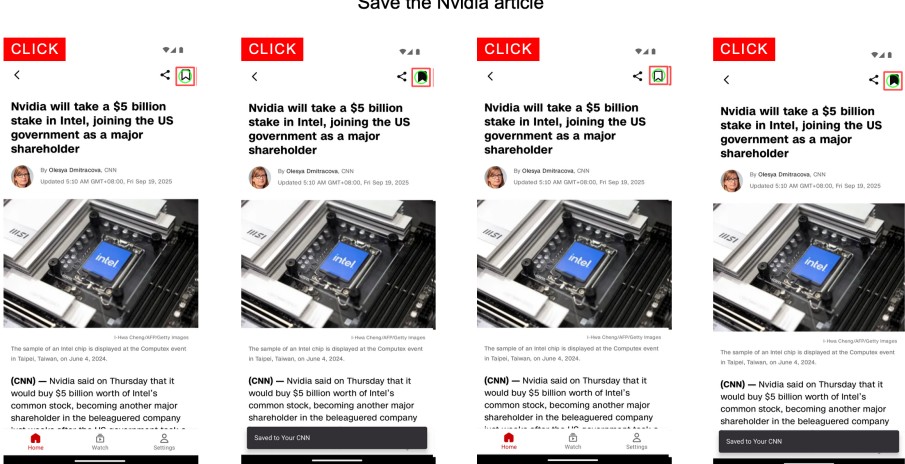

Figure 6: Example for redundant actions.

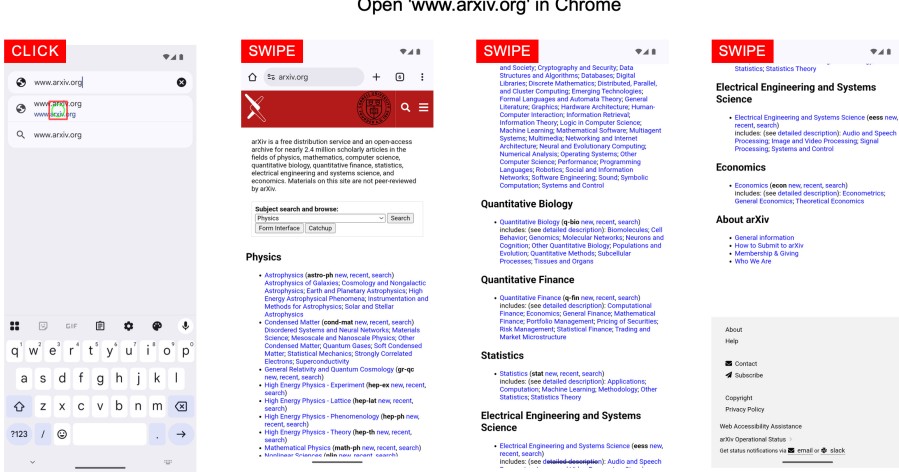

Figure 7: Example for failure to terminate.

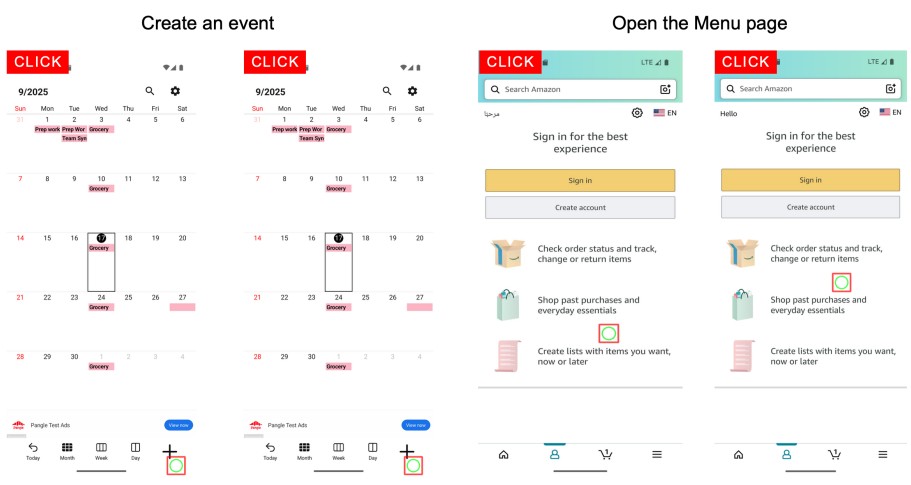

Figure 8: Examples for screen misunderstanding.

