# OpenReview forum: "A3: Android Agent Arena For Mobile GUI Agents"
_ICLR.cc/2026/Conference — ICLR 2026 Conference Withdrawn Submission_

### Official Review · Reviewer_DsBE · 2025-10-31

**Soundness:** 1
**Presentation:** 3
**Contribution:** 2
**Rating:** 2
**Confidence:** 4

**Summary:**

This paper introduces **Android Agent Arena (A3)** — a new benchmark and evaluation framework for **mobile GUI agents**. It proposes an *essential-state* evaluation metric, where large multimodal models (MLLMs) such as Gemini-2.5-pro or the fine-tuned *A3RM* model act as evaluators to assess whether critical intermediate “essential states” are achieved during task execution. The benchmark includes 100 tasks across 20 online apps, and the authors release a full toolchain for agent-device interaction and trajectory evaluation.

**Strengths:**

1. The goal of handling dynamic, real-world app environments is timely and relevant.
2. The paper provides an open-source evaluation toolkit and reward model that could be useful for replication.

**Weaknesses:**

1. **Unconvincing evaluation results:**

    The A3 results **contradict** established model rankings from prior benchmarks. For example, *UI-TARS-1.5* and *Qwen2.5-VL* perform nearly the same here, despite large known differences in their reasoning and visual grounding capabilities. Meanwhile, *InfiGUI-R1* — a less data-rich model — inexplicably outperforms all others. These inconsistencies suggest that **A3’s evaluation method may not reflect real model competence**, but noise from MLLM judgment or benchmark artifacts.

2. **High inconsistency with the authoritative benchmark AndroidWorld:**

    The results on A3 are **highly inconsistent** with those on the authoritative benchmark *AndroidWorld*. Even though *AndroidWorld* uses what the authors describe as a “static” evaluation approach, its **rule-based validation produces convincing and reproducible results** that reliably reflect model capability. For example, *UI-Venus* achieves **49%[1]**, *GUI-Owl* **66.4%**[2]**,** and *UI-TARS-1.5* **64.2% [3]** on AndroidWorld — a ranking that aligns well with their expected strengths. However, in this paper, the relative order of these models changes dramatically. This discrepancy **casts serious doubt on the credibility and validity of A3’s evaluation methodology**, suggesting that its MLLM-based judging process may not correlate with genuine agent competence

3. **No human validation:**

    The claim that A3RM is “human-aligned” is unsubstantiated without **comparison with human evaluators.** Since A3RM is fine-tuned partially using Gemini’s evaluations, comparing A3RM against Gemini and finding alignment could simply indicate **self-consistency**, not real-world validity.

4. **Speculative analysis without ablation:**

    The statement that “the capabilities of the base model appear to be a dominant factor” is **not experimentally verified**. Without controlled experiments isolating model size, architecture, and training data, such claims are anecdotal.


[1] Gu, Zhangxuan, Zhengwen Zeng, Zhenyu Xu, et al. “UI-Venus Technical Report: Building High-Performance UI Agents with RFT.” arXiv:2508.10833. Preprint, arXiv, August 14, 2025. https://doi.org/10.48550/arXiv.2508.10833.

[2] Ye, Jiabo, Xi Zhang, Haiyang Xu, et al. “Mobile-Agent-v3: Fundamental Agents for GUI Automation.” arXiv:2508.15144. Preprint, arXiv, August 21, 2025. https://doi.org/10.48550/arXiv.2508.15144.

[3] UI-TARS-1.5-7B https://seed-tars.com/1.5/

**Questions:**

1. Can the authors **quantitatively validate** MLLM evaluation accuracy against human annotators on a subset of tasks?
2. How do they explain the **inverted ranking** of models relative to AndroidWorld? Is there any evidence that A3 better reflects human judgment rather than being more random?
3. Could the authors report **inter-rater agreement metrics** (e.g., Cohen’s κ) between A3RM, Gemini, and humans?

---

> ### Author Response · Authors · 2025-11-18
> **Response to reviewer DsBE**
>
> Thank you for constructive feedback and insightful suggestions for improvement.
>
> **Weakness**
>
> 1. We have updated our reward model A3RM and the evaluation results in **(Part 1) For all reviewers**. However, the updated results still show a similar performance rankings. Though you think the rankings are doubtable, we find the results interesting. UI-TARS or GUI-Owl performs good on AndroidWorld, but in the real world testing, they indeed work strangely. Ui-TARS suffers from repeating same wrong action even with history data while GUI-OWL has bad plan capability and weird behavior even on AndroidWorld (output random input text instead of the required one, which may seem overfit to training data). Besides, the gap between AndroidWorld and A3 is huge. Think of the tasks in AndroidWorld: create notifications, start voice recording (which can actually be completed by API-based mobile assistants such as Siri, why test GUI agents on these tasks), open a customized html and draw a painting (who would ask GUI agents to do that). And A3 provides tasks such as summarizing news, finding the cheapest flight ticket, playing a video when we are cooking, searching for products in shopping apps and want to know the price, or checking the navigation when you start driving. The high performance on AndroidWorld does not imply high performance on A3. On the contrary, InfiGUI has low scores on AndroidWorld, but it show strong generalization on realistic tasks on A3, which is reasonable. The rankings on A3, however, reveals that agents perform differently on different benchmarks, which is reasonable.
> 2. Same as 1.
> 3. In section 4.2 line 387 and Appendix A.4, we only use 11% of the training data from Gemini labeling, and they are only negative samples as the Gemini has high recall but low prevision. Most training data (89%) are from human collection and annotations. In the old paper, Table 3 shows the evaluation results of A3RM from human verification. However, we have updated A3RM and now it achieves higher evaluation accuracy on the test data, which is collected from real agent data and annotated by human (see **(Part 1) For all reviewers**). The table shows that the judgment by A3RM is well aligned to human justification. The idea of using reward model is to reduce the human labor in agent evaluation and since our model has passed the test data, we believe human evaluation is not needed to verify all the eight agents trajectories.
> 4. Thanks for the suggestion. We experimented with T3A using Qwen2.5VL-7B as the base model for ablation. The results are in the **(Part 1) For all reviewers**. The huge performance gap between Qwen2.5VL-7B and Gemini shows that the base model is the main factor. While Mobileuse also uses Qwen2.5VL-7B as the base model, it achieves higher performance than T3A, which shows that the framework design is also important.
>
> **Question**
>
> 1. Table 3 in the old paper is the evaluation of MLLM evaluation accuracy. The new version is in the **(Part 1) For all reviewers**.
> 2. Same as weakness 1.
> 3. Cohen's $\kappa$ is a suitable metric to measure the agreement between evaluators. However, we now only have the observed agreement and not have a good measure for the expected agreement for these evaluators. We are happy to have a discussion on this.

---

### Official Review · Reviewer_sVzZ · 2025-11-01

**Soundness:** 3
**Presentation:** 3
**Contribution:** 2
**Rating:** 2
**Confidence:** 3

**Summary:**

This work presents a novel benchmark on GUI agents for mobile device control, addressing the drawbacks of existing prior benchmarks. The proposed benchmark features the use of online apps, unlike other benchmarks that have focused mostly on static apps, and a novel evaluation system that can benefit from scalability and dense signals. The authors also rigorously investigate the effectiveness of the proposed evaluation system. By evaluating many state-of-the-art GUI agents on the benchmark, the authors reveal that significant challenges exist towards effective in-the-wild agents.

**Strengths:**

This work presents several strengths, which I detail below:
1. Evaluation system: The authors present a novel mechanism to evaluate agents in dynamic, online applications. The introduced evaluation method has two features: incorporation of critical states, use of MLLM, which I believe is highly beneficial in many cases. By using critical states, the users/agents can get more fine-grained signals upon success. Also, using MLLM allows the evaluation design to be scalable without necessitating human experts.
2. Comprehensive results: Results across diverse agents are presented, which potentially provide a meaningful reference point for future work.

**Weaknesses:**

Above all, while this work presents noticeable strengths, the main weakness lies in the ambiguity of the contribution compared to existing work. I list specific weaknesses/questions/suggestions regarding this work.
1. Motivation: While the authors emphasize that this work allows evaluation of the online apps, the reason behind why we need to evaluate online apps is not presented thoroughly. I am not fully convinced whether such a feature is highly demanded by the community. Gemini-2.5-pro, achieving around 55 task success rates, also reveals that this benchmark does not provide enough headroom for improvements. More elaboration on what the users can benefit from by evaluating this benchmark (as well as online apps) should be provided.
2. Specification of essential state: The detailed explanations of the process for defining essential states for each task are missing. What criteria are used to define the number and content of essential states for each task?
3. Use of essential-states for RL: While it is out of scope as a benchmark, as it is currently illustrated, the dense signal of essential-state evaluation can be useful for RL. I do think that investigating AR3M to RL and comparing with existing reward mechanisms can be a valuable study for the community, as adopting RL for GUI agents is gaining interest [1,2].
4. More analysis on the agents' behaviors: Although the authors present analysis of the agents' behavior in Appendix 5.2, the observations are already known to the community (very similar observations are already present in the work presented in the related work; for example, the lack of ability to accurately ground the intended action is widely studied [3,4]). As a reader, I want to know more about failure modes strongly correlated to the focus of this paper (e.g., the dynamic nature of online apps).

---

References

[1] “Digi-q: Learning q-value functions for training device-control agents.”

[2] “UI-TARS: Pioneering Automated GUI Interaction with Native Agents”

[3] “GPT-4V(ision) is a Generalist Web Agent, if Grounded”

[4] “GUI-Bee: Align GUI Action Grounding to Novel Environments via Autonomous Exploration”

**Questions:**

(Questions/suggestions are included in the above section)

---

> ### Author Response · Authors · 2025-11-18
> **Response to reviewer sVzZ**
>
> Thank you for constructive feedback and insightful suggestions for improvement.
>
> **Weakness**
>
> 1. Previous benchmarks, especially mostly used AndroidWorld and AndroidLab, only use offline apps, which does not provide a suitable benchmark for GUI agent. They only provide apps and tasks such as create notifications, start voice recording (which can actually be completed by API-based mobile assistants such as Siri, why test GUI agents on these tasks), open a customized html and draw a painting (who would ask GUI agents to do that). And what we want the agent to actually do such as summarizing news, finding the cheapest flight ticket, playing a tutorial video when we are cooking, searching for products in shopping apps and wanting to know the price, or checking the navigation when you start driving. These tasks are what we expect GUI agents to do and thus we select those online and dynamic apps to test agents' capability on these tasks. Currently the use of general commercial models such as Gemini is indeed the best choice due to the much stronger base capability compared to 7B-ish models. The GUI agent frameworks using general models have achieved 97% on AndroidWorld. 55% SR is not high and there is a lot room for improvements. And the SR for hard tasks is 44% for the latest strongest commercial models. As for the smaller-size models, they can now only achieve under 30% SR, many only around 10%. There is huge room for such smaller-size agents to improve on our tasks.
> 2. See details in **(Part 1) For all reviewers**.
> 3. yes we are developing a general reward model for more robust use cases. But currently it is not the focus of this work.
> 4. In line 419 in section 4.3, the results of our work shows that even though current agents (such as GUI-OWL, UI-TARS, etc) can achieve high scores on offline static benchmarks such as AndroidWorld (73% and 64%), they can only achieve around 10% on A3 tasks, which actually demonstrate more realistic and real-life scenarios. We indeed find this results interesting, since it seems those models may overfit to AndroidWorld. On the contrary, InfiGUI-R1 has much lower scores on AndroidWorld, it achieved much higher scores on our benchmark, which shows biased evaluation of AndroidWorld. Also what we find in Appendix 5.2, we notice the common issue of the existing agent, which is the progress unawareness, this issue is clearly explained in the appendix. Grounding issue is widely studied but we still need to mention it for the comprehension of the error case study. We believe if these two issues are solved, the performance of the agents will be improved significantly.

---

> ### Comment · Reviewer_sVzZ · 2025-11-23
> **Remaining questions on the Author's rebuttal**
>
> Thanks for clarifying and detailing many of the points I asked about. The answers resolved my concerns regarding the essential state. However, I present the remaining concerns below.
>
> 1. I agree that 55% SR is not perfect. However, I can't agree that this benchmark provides anything significantly different from AndroidWorld and related works. The exemplary tasks you provided in the paper (in the Appendix, such as “Save the Nvidia article” or “Open ‘www.arxiv.org’ in Chrome”) do not reflect your goal (such as “summarizing news” or “finding the cheapest flight ticket”). The use of online apps is already present in SPA-Bench (and it provides tasks with ‘Chinese’, which is a unique feature). That said, given the 44% SR on the hard tasks, I suggest updating the tasks to include more hard tasks so the benchmark remains more challenging and aligns with the ultimate goal you provided.
> 2. Thanks for clarifying the protocol. I value the defined metric. Also, I agree that incorporating three human performers alleviates the challenges of defining essential states in nature.
> 3. Thank you for the answer.
> 4. I agree that progress awareness and action grounding are two main bottlenecks of current GUI agents. Yet, I’m more curious about the failure cases specific to the unique feature of your benchmark. To elaborate, SPA-Bench examines the different behaviors of agents using different languages (Appendix F.2 in their paper), and B-Moca presents the different behaviors of agents using different device configurations (Figure 5 in their paper). Given that one of the unique features of this benchmark is the use of online apps, I request that the authors illustrate failure cases related to task characteristics of involving online apps.
>
> I have updated my rating from 2 to 4.

---

> ### Author Response · Authors · 2025-11-24
> **Response to reviewer sVzZ**
>
> Thank you for your quick response and updated rating. For your remaining questions, we would like to further explain
>
> **Question 1**
>
> First, regarding task alignment: We acknowledge that the examples cited in the original paper (e.g., "Save the Nvidia article") did not fully reflect our high-level goals. We have addressed this by providing ten additional task examples in the **(Part 2) For All Reviewers** response, which demonstrate better alignment with goals. However, we note that task design is constrained by the inherent functionality of specific apps (e.g., CNN is strictly for news reading, YouTube for video playing). To create diverse tasks within a single app, we have to vary the format, which necessitates simpler objectives like "Open arxiv in chrome."
>
> Second, regarding the "Hard" subset and Success Rate (SR): We respectfully argue that the current difficulty level is appropriate for the agents. While Gemini achieves 44% on hard tasks, smaller, on-device models (e.g., 7B parameters) typically achieve less than 20%. This gap highlights substantial room for improvement of small size models, since the ultimate goal of mobile GUI agents is on-device deployment, which ensures user privacy and reduce cost. we view Gemini’s performance as a "lighthouse" or upper bound for future research, rather than a baseline that implies the benchmark is too easy.
>
> **Question 4**
>
> Thank you for the clarification. We have analyzed failure cases specific to the unique, dynamic nature of online apps and identified three primary categories of failure:
> 1. Information Similarity & Ranking: Online apps often present search results where the correct item is buried amidst highly similar distractions. Agents may get confused by the similar entries displayed.
>   - Example: In YouTube, when searching for "How to cook steak" by a specific uploader, the target video may require scrolling twice. Agents often fail to scroll and select the top (incorrect) video instead, which has a similar title.
> 2. Dynamic Interference (Ads & Pop-ups): Unlike static offline apps, online apps (e.g., Joytify, Home Workout) feature unpredictable pop-up ads or sponsored content.
>   - Example: When searching for "NYC rent" in Chrome, the top results are often sponsored links. Agents must distinguish between organic and sponsored results to find the specific information requested. Similarly, agents frequently fail to identify and close sudden pop-up ads (either in-page or whole-screen), causing task failure.
> 3. Rich information and layout design: Many apps provide rich information of the search item and many apps have deep and wide hierarch design, which makes the agent planning harder.
>   - Example: Amazon has much information for a product, and if we want the agent to check the product details, it may be confused where to find the details. And filtering and sort are also misleading. In Amazon and Omio, where tasks require agents to sort and filter, there exists many filters and sorting options even with second-level options, making agents hard to decide what to do.

---

### Official Review · Reviewer_XKrY · 2025-11-01

**Soundness:** 3
**Presentation:** 3
**Contribution:** 2
**Rating:** 4
**Confidence:** 3

**Summary:**

This paper introduces a new benchmark and evaluation framework for testing mobile GUI agents in realistic, dynamic environments. Unlike prior benchmarks that rely on static frames or offline apps, A3 includes 100 tasks across various real-world online applications from the Google Play Store, reflecting real-world user interactions. It proposes a novel “essential-state” evaluation method that uses commercial or open-source multimodal LLMs (such as Gemini or A3RM) as automated evaluators to verify task progress and completion with reduced manual effort.

**Strengths:**

- Evaluates existing GUI agents on a broader and more practical set of real-world applications compared to prior benchmarks.
- Proposes the essential-state evaluation technique, enabling more accurate assessment of intermediate task success where traditional functional-based approaches often fail to capture partial progress.

**Weaknesses:**

- Using closed-source and widely used commercial applications is not, by itself, a novel contribution. Previous benchmarks such as B-MoCA and AndroidLab have also incorporated commercial apps, and similar extensions could be made by simply adding new tasks.
- The paper does not address cases where MLLM-based success detection may be inaccurate. Furthermore, even if deterministic settings are enforced, MLLMs inherently exhibit stochasticity, meaning success judgments could vary across evaluations. It would strengthen the paper to discuss mitigation strategies for this issue.
- While the authors claim that essential-state evaluation overcomes the limitations of functional-based methods, the paper would benefit from a case study clearly illustrating scenarios where functional-based methods fail but essential-state evaluation succeeds.
- The experiments compare only zero-shot performance of publicly available GUI agents. It would be interesting to see results from BC (behavioral cloning) or RL fine-tuning on A3 tasks to evaluate learning improvements.
- The method for defining essential states is described ambiguously, and the full list of essential states for all benchmark tasks is not made publicly available.

**Questions:**

- Are there any copyright or licensing concerns related to using private company applications in the benchmark?
- Is there a generalizable rule or guideline for defining “essential states”? When adding new tasks to the benchmark, do these states have to be manually specified by humans using heuristics?

---

> ### Author Response · Authors · 2025-11-18
> **Response to reviewer XKrY**
>
> Thank you for constructive feedback and insightful suggestions for improvement.
>
> **Weakness**
>
> 1. We would like to claim that the contribution is from both the widely-used aspect and the dynamic aspect of the apps we select. AndroidLab and B-Moca does not provide dynamic app (content refreshing) such as news, ticketing, shopping, etc. However, we think that a general GUI agent should be able to do such tasks on those popular apps. It is very common for users to ask the agent to summarize news when wake up or to play a tutorial video when cooking. We are providing such tasks and apps that can test such capability of agents. We believe that is a critical contribution.
> 2. We have updated our reward model A3RM and it is now accurate to serve as a evaluator. We think the error rate is acceptable since from current benchmark results, 4% in ESAR and 2% in task SR does not show strong improvement in the dynamic environment. Besides, after finetuning, the outputs of A3RM remain same across evaluation runs and thus the performance stochasticity is not an issue.
> 3. Functional-based methods fail in two key areas where essential-state evaluation succeeds. First, they cannot evaluate information-query tasks especially when the apps are dynamic. For example, in a task like "Find the top recommended activity in Manchester on TripAdvisor," a functional method can not get the ground truth of a dynamic content. Our essential-state method is designed to validate this specific information query by VLM capability. Second, functional methods struggle with partial completion and flexible trajectories. They often require a specific path or final state and register a binary pass/fail. If an agent completes half the task via an unexpected route, it's marked as a failure. Our method evaluates the states achieved, allowing us to measure partial success. This is why we introduced the ESAR metric, which provides a more granular comparison than task-level success rates, especially when two models appear to perform similarly.
> 4. this is indeed interesting, but we think that's not our main focus for this work.
> 5. We provide details of essential states in **(Part 1) For all reviewers**.
>
> **Question**
>
> 1. We only provide a device image which contains the apps. If researchers want to evaluate the models, they need to login with their own accounts. Besides, we do not collect any data or crawl any data from the apps. The project is only made for academic research.
> 2. The first part can be found in **(Part 1) For all reviewers**. FOr the second question, the answer is yes. If new tasks are added, users need to create new essential states because each task is different and the essential states are different.

---

> ### Comment · Reviewer_XKrY · 2025-11-24
>
> Thanks for your response. I have a few remaining questions.\
> [W1] How does it differ to incorporate such dynamic apps into AndroidLab and B-Moca? Are there any specific challenges and your novel solution in that regard?\
> [Q1] Do you mean your evaluation setup is not based on running simulators, or is it just using static, pre-captured images?\
> [Q2] I have some concerns about the scalability of the benchmark, as defining essential state is highly heuristic, and if the task horizon becomes much longer, there could be multiple ways to complete the entire task, while the provided tasks remain relatively simple.
>
> As a result, I would like to retain my score.

---

> ### Author Response · Authors · 2025-11-24
> **Response to reviewer XKrY**
>
> **[W1]**
>
> First, B-Moca and AndroidLab they both use function-based evaluation, the entire codebase is designed for function evaluation, which is completely different from our evaluation method. What we design is an interaction toolkit and another separated evaluator. The toolkit focuses on the interaction and data collection, which means without the evaluation system, given any task, users or agents can use this toolkit to collect trajectories with or without essential states. This toolkit depends only on the Android SDK platform-tools, which is required to install when using the AVD. Thus the environment setup is extremely simple and all the evaluation is done after the data trajectories are collect. There is no simple way to directly incorporate our apps and tasks to their codebase and they do not provide instructions of customized agents or tasks/apps. If too much code refinement or implementation is required, why not directly build a more flexible and lightweight toolkit for the community to use. However, we admit this toolkit contribution is a engineering solution, which does not imply engineering novelty, it is only because we want to develop a more easy-to-use toolkit. Our main novelty lies in the apps / tasks and VLM-based essential state evaluation method.
>
> **[Q1]**
>
> Sorry for the confusion of the word "image". The entire system is working on Android devices (either AVD or real device). When I said we provide a device image, we mean a device state snapshot (not screenshots), which contains app information and state information for a reproducible reset and initial state.
>
> **[Q2]**
>
> Yes the essential state definition is heuristic and all based on humans, since we have tested the capability of the latest LLMs to generate but they can hardly output precise essential states. In the **(Part 2) for all reviewers**, we have included hard tasks such as "Find the cheapest flight from Paris to Rome for 2 adults on Omio departing tomorrow. What is the total price?". This task has 7 essential states and requires more than 15 steps even for human to complete. If you take a look at the essential states, none of them can be dropped and they are "common states" even if multiple ways exist to finish the whole task. If any of the essential state is not achieved, the task cannot be complete as required. One agent can first set the destination to Rome then set the departure city to Paris, another agent can first set the departure date then set the destination. But in common, they all have to set these states. The scalability requires humans to create tasks and manually set the essential states following the principles we mentioned in **(Part 1) for all reviewers**.

---

### Official Review · Reviewer_tFAx · 2025-11-01

**Soundness:** 3
**Presentation:** 3
**Contribution:** 2
**Rating:** 6
**Confidence:** 4

**Summary:**

The paper proposes A3, a dynamic benchmark for mobile GUI agents built on 100 human-designed tasks from 20 popular, online Android apps spanning 20 categories (news, travel, shopping, fitness, notes, etc.). Unlike prior dynamic benchmarks that rely on open-source/offline apps (AndroidWorld, AndroidLab), A3 deliberately targets real, changing, store apps that users actually use, and therefore cannot be instrumented for function-based checking. To make such apps evaluable, the authors introduce an essential-state evaluation: each task is decomposed into several key milestones (e.g. “search X”, “open first item”, “answer author”), and a MLLM-based judge (Gemini or a distilled model A3RM) checks, via a sliding window over the trajectory, whether each essential state has been achieved. This yields both a normal SR and a finer ESAR (Essential-State Achieved Rate), which exposes partial progress on long tasks. They also release an execution toolkit (AITK) and an evaluator module. On eight mobile agents, SR is low (∼30% for the best open models; only T3A+Gemini gets ~58%), but ESAR is much higher, showing current agents often “get halfway” but fail late.

**Strengths:**

- Ecological validity: using real, online apps that update and may show ads/popups — exactly the scenarios offline/open-source benchmarks cannot cover. This is the main novelty claim.
- Semantically aware evaluation: essential-state + sliding-window MLLM judging is a clearer and more reliable formulation than “judge the whole multi-step trajectory in one go,” which the authors note SPA-bench’s MLLM-style judging does only at ~80% accuracy. A3 breaks the task so the judge reasons locally.
- Cost-aware judging: commercial MLLMs work best, but the paper actually trains A3RM (MiMo-VL-7B finetuned) to get close to Gemini-level essential-state accuracy and make the benchmark practical.

**Weaknesses:**

- MLLM-as-judge brittleness remains: although essential-state simplifies judging, Sec. A.5 still shows hallucination and semantic misalignment (judge thinks “send” worked, or confuses “title” vs “cost”). This means reproducibility across model versions / prompts is still an issue.

**Questions:**

See weakness

---

> ### Author Response · Authors · 2025-11-18
> **Response to reviewer tFAx**
>
> Thank you for the positive rating and the insightful suggestion.
>
> **Weakness**
>
> 1. We updated our reward model A3RM and now it shows much less hallucination and the evaluation results are more stable and reproducible (see **(Part 1) For all reviewers**).

---

### Official Review · Reviewer_y3j8 · 2025-11-05

**Soundness:** 2
**Presentation:** 3
**Contribution:** 2
**Rating:** 4
**Confidence:** 4

**Summary:**

This paper introduces Android Agent Arena (A3), a benchmark system for evaluating mobile GUI agents. The main contributions include: (1) A benchmark of 100 tasks spanning 20 popular online mobile applications across diverse categories from the Google Play Store, (2) A novel "essential-state" evaluation methodology that decomposes tasks into verifiable intermediate milestones and uses MLLMs as automated judges, (3) A3RM, a fine-tuned 7B reward model that provides cost-effective evaluation compared to commercial APIs, and (4) An open-source pipeline (AITK) for agent execution and data collection. The benchmark addresses limitations of prior work by including dynamic, online applications (e.g., shopping, news, travel) that were previously untestable due to reliance on open-source apps. Evaluation of 8 mobile GUI agents reveals that A3 poses significant challenges, with the best agent achieving only 29% success rate under commercial MLLM evaluation.

**Strengths:**

1. The inclusion of online, commercial apps from diverse categories (shopping, travel, news) fills a critical gap in existing benchmarks that rely on offline/open-source apps.

2. The essential-state decomposition provides interpretable, fine-grained metrics (ESAR) beyond binary success rates, enabling better diagnosis of agent capabilities.

3. The paper's strongest point is its focus on "in-the-wild" evaluation. By using 20 popular, dynamic, online apps , A3 presents a more realistic and challenging environment than benchmarks restricted to offline or open-source apps , which cannot test robustness to dynamic content or real-world UI.

**Weaknesses:**

1. Despite improvements, A3RM still has ~10% error rate (89.9% F1). The paper documents persistent failure modes (visual hallucination, semantic misinterpretation) but doesn't adequately address whether this error rate is acceptable for benchmark validity. More critically, there's no analysis of whether these errors systematically favor certain agents.

2. Essential states are manually created without inter-annotator agreement metrics or user studies validating that they appropriately decompose tasks. The claim that they provide "uniform granularity" lacks empirical support.

3. The paper claims A3 is superior to SPA-bench because A3 allows for "reliable and programmatic resets" while SPA-bench suffers from "severe environmental instability and difficult resets". This is a critical claim, as SPA-bench also uses online apps. However, this claim is never substantiated. The paper provides no details on its "programmatic reset" mechanism for dynamic apps nor any direct comparison to prove A3 is more stable. Without this, A3's novelty over SPA-bench is unclear.

4. The entire framework hinges on the quality of the manually defined "essential-states". The process for defining these is opaque. The paper states they were created to have "uniform granularity", but this seems subjective. The work would be much stronger if it detailed the rubric or principles for defining these states and how consistency was maintained across 100 tasks.

5. While API costs are mentioned, there's no full cost-benefit analysis comparing manual evaluation, function-based evaluation, and MLLM-based evaluation across dimensions like accuracy, development time, and maintenance burden.

6. A3RM is trained on Gemini labels and validated against Gemini, potentially inheriting and amplifying Gemini's biases rather than providing an independent evaluation standard.

**Questions:**

1. How were essential states validated? What was the inter-annotator agreement? Have you conducted user studies to verify that the essential-state decompositions align with human perception of task progress?

2. What is your plan for maintaining the benchmark as apps update? Will you version-lock apps, continuously update tasks, or periodically release new benchmark versions?

3. Given the ~10% error rate for the MLLM judges, how confident can we be in the agent rankings in Table 4? Have you considered quantifying this uncertainty, perhaps by reporting a confidence interval on the SR and ESAR scores based on the evaluator's known F1 score?

4. How sensitive are results to the granularity of essential-state decomposition? What happens if you use coarser or finer decompositions?

5. What is human performance on this benchmark? This would provide crucial context for interpreting agent scores.

6. Have you measured how frequently the included apps update in ways that break tasks? What percentage of tasks remain valid after 3/6/12 months?

7. You state A3RM is "stricter" than Gemini , but it also uses a smaller context window (2 frames vs. 4). How do you disentangle these two factors? Is it possible A3RM is not "stricter" but simply less accurate due to its limited context, causing it to miss valid achievements that Gemini catches?

---

> ### Author Response · Authors · 2025-11-18
> **Response to reviewer y3j8**
>
> Thank you for constructive feedback and insightful suggestions for improvement.
>
> **Weakness**
>
> 1. In the **(Part 1) For all reviewers**, we have retrained and updated A3RM and now it only has around 4% essential state evaluation error rate and 2% task trajectory evaluation error rate. We think the error rate is acceptable since from current benchmark results, 4% in ESAR and 2% in task SR does not show strong improvement in the dynamic environment. Besides, The test data are from diverse agent collection and thus it does not show bias.
> 2. Details of essential states creation verification is also in the **(Part 1) For all reviewers**.
> 3. This is basically a programmatic engineering solution plus our app/task selection. SPA-bench claimed to provide a stable and reproducible android emulator snapshot but it never releases. And the apps/tasks it selects requires human reset and their codebase does not provide "program reset", which causes extensive manual labor to reproduce. Instead, we are providing a codebase that can easily use predefined snapshot and images to reset the device to the initial state for reproducibility.
> 4. same as 2
> 5. We believe manual evaluation and function-based evaluation are not in the consideration for our dynamic benchmark. Manual evaluation is surely not applicable and extensible especially for multiple evaluation tries. As for the function-based evaluation, we admit that function-based evaluation has possibly the highest accuracy and the cost is one-time (you need to define one function for each task when building the benchmark), however, function-based evaluation can't evaluate dynamic apps and tasks. The contents of those apps are refreshing and many tasks can be achieved through multiple ways and functions can't catch all these situations. Therefore we have to use MLLM-based evaluation for our tasks and apps. And we only compare the cost for general API since we assume that our reward model (7B) is deployable on server and that cost much less than the API.
> 6. In Section 4.2 and Appendix A.4, the training data are most from human collection (see **(Part 1) For all reviewers**) and Gemini labeled data only provides 11% negative samples. The small number of Gemini labeled data do not affect the training when it can mostly successfully label the negative samples.
>
> **Question**
>
> 1. More details are in **(Part 1) For all reviewers**
> 2. We will provide a device image to lock the app versions and thus the interfaces do not change.
> 3. We can not compute the confidence interval from the F1 score of the evaluator due to statistical nature, but since we have updated A3RM, the confidence interval is not that important
> 4. We ensure the essential states to be comprehensive and essential states are carefully designed. It is hard to try different granularity of the decomposition. But we think coarser decomposition will cause that the essential states can't be judged within two steps. In most case, decomposition cannot be finer since current essential states are common states or actions from distinct trajectories, finer essential states will lead to misjudgment of multi-way of achieving current essential states.
> 5. Human can do 100% of the tasks. We ensure that all tasks can be completed by humans and we collected 3 distinct trajectories for each task.
> 6. Since we lock the app versions using the device image, the app updates or task maintenance is not a concern.
> 7. As we updated our reward model, this claim is more stronger since we notice the "strict" evaluation. Also, we study the sliding window size and Gemini has similar evaluation performance when windows size is 2, 3 and 4. We select 4 because it costs much less. We also ensure that essential states can be judged within 2 steps and the context window size is not the concern.

---

### Author Response · Authors · 2025-11-18
**(Part 1) For All Reviewers: A3RM updates and new experiment results; Detailed explanation of essential states**

We thank the reviewers for their positive feedback and insightful suggestions for improvement. This comment mostly discuss two aspects: **(1) A3RM updates** and **(2) details of essential states**.

## A3RM updates

We also notice the high error rate of the original version of A3RM and to address the issue, we have retrained and updated A3RM and now it is advanced (comparable to Table 3) and surpass the performance of Gemini and is well-aligned to human judgment by around 4% essential state error rate and 2% trajectory error rate. To be specific, this evaluation is tested on real agent trajectories and human annotations as ground truth.

| Model | Ess. State Precision  | Ess. State Recall  | Ess. State F1  | Ess. State Accuracy | Task Precision  | Task Recall  | Task F1  | Task Accuracy |
| :---: | :---: | :---: | :---: | :---: | :---: | :---: | :---: | :---: |
| Gemini-2.5-pro | 87.3 | 96.3 | 91.5 | 89.5 | 85.7 | 96.0 | 90.6 | 95.0 |
| A3RM (old) | 89.2 | 90.0 | 89.6 | 89.5 | 91.7 | 88.0 | 89.9 | 95.0 |
| A3RM (new) | 95.7 | 94.8 | 95.3 | 96.6 | 96.0 | 96.0 | 96.0 | 98.0 |

Using the new A3RM as the evaluator, we re-evaluate the benchmarks as in the following table:

| Agent | Metric | Easy | Medium | Hard | Operation | Inf. Query | Overall |
| :---: | :---: | :---: | :---: | :---: | :---: | :---: | :---: |
| UI-TARS-1.5 | SR | 14.3 | 2.5 | 0.0 | 8.3 | 0.0 | 6.0 |
|  | ESAR | 28.2 | 15.6 | 10.4 | 19.2 | 13.2 | 17.5 |
| UI-Venus | SR | 31.4 | 12.5 | 16.0 | 19.4 | 21.4 | 20.0 |
|  | ESAR | 43.5 | 31.3 | 27.1 | 36.7 | 25.3 | 33.3 |
| UI-Genie | SR | 28.5 | 12.5 | 4.0 | 20.8 | 3.6 | 16.0 |
|  | ESAR | 44.7 | 25.0 | 34.3 | 34.8 | 29.7 | 33.3 |
| Qwen2.5-VL | SR | 11.4 | 2.5 | 0.0 | 6.9 | 0.0 | 5.0 |
|  | ESAR | 14.1 | 11.7 | 20.8 | 17.4 | 9.9 | 15.2 |
| InfiGUI-R1 | SR | 37.1 | 30.0 | 12.0 | 33.3 | 14.3 | 28.0 |
|  | ESAR | 61.1 | 54.6 | 52.1 | 55.5 | 54.9 | 55.3 |
| GUI-OWL | SR | 25.7 | 5.0 | 0.0 | 13.9 | 3.6 | 11.0 |
|  | ESAR | 49.4 | 26.5 | 23.9 | 35.8 | 23.1 | 32.1 |
| MobileUse+Qwen2.5VL-7B | SR | 37.1 | 7.5 | 4.0 | 22.2 | 3.6 | 19.0 |
|  | ESAR | 49.4 | 41.4 | 31.3 | 42.7 | 35.2 | 40.4 |
| T3A+Gemini-2.5-pro | SR | 60.0 | 57.5 | 44.0 | 58.3 | 46.4 | 55.0 |
|  | ESAR | 70.6 | 69.5 | 63.5 | 74.7 | 51.6 | 67.9 |
| T3A+Qwen2.5-7B | SR | 37.6 | 28.1 | 28.2 | 29.3 | 34.1 | 30.7 |
|  | ESAR | 31.4 | 10.0 | 4.0 | 19.4 | 3.6 | 15.0 |

From reviewer DsBE, we also include a new try using T3A with Qwen2.5VL-7B as the base model for ablation.

## Essential State Details

We apologize that the essential state creation was not sufficiently detailed. We will revise the paper to explicitly describe our multi-stage, human-in-the-loop procedure for designing and verifying the essential states. To be specific, we defined simple but effective criteria for this process:
- The state must be clearly identifiable from two consecutive screenshots and the intervening action.
- The state must represent a critical "must-be-done" step for a successful task completion.
- The set of all essential states must be sufficient to cover the entire task.

However, we admit that a purely quantitative evaluation of the "quality" of essential state design is challenging, as the process is inherently about defining semantic milestones in a human-centric task. But, we ensured robustness and mitigates individual-operator bias through:
1. Two human operators are instructed to complete the same task in distinct ways and collected various successful trajectories.
2. Based on the collected data, these two operators collaboratively discussed and proposed a set of "essential states." This ensures the states are general enough to cover different valid strategies (e.g., states or actions that are common even when trajectories differ).
3. A third human, who did not participate in the initial design, was then asked to verify the full set of essential states for each task to ensure they were logical, achievable, and comprehensive.

To provide a clearer and more persuasive demonstration, a list of essential states are in the next part.

---

> ### Author Response · Authors · 2025-11-18
> **(Part 2) For all reviewers: task and essential states list**
>
> We provide 10 tasks from different apps and their corresponding essential states:
> - Navigate to CNN's Science section and check the top headline news. What is the title?
>   - CNN Science section is selected
>   - The top headline news in Science section is selected
>   - The title of the article is answered
> - Search for 'hotels in Seoul' on Tripadvisor for 1 room and 2 adults, sorted by traveler ranking.
>   - Room is set to 1 room
>   - Guest details are set to 2 adults
>   - 'hotels in Seoul' is searched
>   - Results are sorted by traveler ranking
> - Open Amazon and search for 'Laptop Sleeve'. Filter the material by 'carbon fiber'. What is the price of the first result?
>   - 'Laptop Sleeve' is searched
>   - Results are filtered by material 'carbon fiber'
>   - Price of the first result is answered
> - Find the cheapest flight from Paris to Rome for 2 adults on Omio departing tomorrow. What is the total price?
>   - Departure location is set to Paris
>   - Arrival location is set to Rome
>   - Passengers set to 2 adults
>   - Departure date set to tomorrow
>   - Flights are searched
>   - Results for flights are sorted by 'Cheapest price'
>   - Cheapest flight price is answered
> - Open Citymapper and search route from London Bridge to Oxford Street. How long is the estimated walking time?
>   - Route starting location is set to London Bridge
>   - Route destination is set to Oxford Street
>   - Estimated walking time is answered
> - Open Gmail and send an email to 'stock_notify_01@163.com' with subject 'Meeting time' and body 'When is the meeting?'
>   - New email page is opened
>   - Recipient set to 'stock_notify_01@163.com'
>   - Subject 'Meeting time' is added
>   - Body 'When is the meeting?' is added
>   - Email is sent
> - Search 'How to cook steak' in Youtube, play video by 'Hodder Books' titled 'Gordon Ramsay’s Ultimate Cookery Course'
>   - 'How to cook steak' is searched
>   - Video by 'Hodder Books' with title 'Gordon Ramsay’s Ultimate Cookery Course' is selected
>   - Video is played
> - Open Home Workout, search Abs workouts, select 'abs beginner', and start training.
>   - Abs workouts are searched
>   - Workout 'abs beginner' is selected
>   - Training is started
> - Remove butter from shopping list in supercook, mark it as pantry ingredient, and tell me how many recipes can be made.
>   - Shopping list is accessed
>   - Butter is removed from shopping list
>   - Butter is marked as pantry ingredient
>   - Number of recipes possible is answered
> - Search 'Quantum Computing' in Wikipedia, select first article, open table of contents, and go to Algorithms chapter.
>   - 'Quantum Computing' is searched
>   - First article is selected
>   - Table of contents is opened
>   - Algorithms chapter is accessed

---

### Note · Authors · 2025-12-04

I have read and agree with the venue's withdrawal policy on behalf of myself and my co-authors.